# Challenges Faced by Multi-Campus Institutions with Online Teaching during the COVID-19 Lockdown

**Covanni Hohls-du Preez** *[ID] **and Ben Marx**

Department of Accountancy, College of Business Economics, University of Johannesburg, Johannesburg 2092, South Africa; benm@uj.ac.za
* Correspondence: chohls@uj.ac.za

**Abstract:** In December 2019, the COVID-19 virus was first detected in Wuhan, Hebei Province, China. On 11 March 2020, the World Health Organisation declared the COVID-19 virus a world pandemic. Even though the physical closure of tertiary institutions had proven to be an effective strategy in breaking the transmission chain in the pandemic, the closure still had various negative effects on students' academic endeavours. In an attempt to keep the disruption of studies to a minimum, higher education institutions across the world implemented online learning programmes by converting face-to-face programmes, thereby creating various challenges for students and lecturers on different levels. The purpose of this article was to determine how challenges faced by multi-campus universities in South Africa during this time would compare to challenges faced by other countries. This study focused on two specific multi-campus universities in South Africa with similar student bodies that offered various accounting courses on different campuses. These courses ranged from diplomas to CTA degrees.

**Keywords:** academic challenges; COVID-19; challenges; online learning; tertiary level

## 1. Introduction

In December 2019, the COVID-19 virus was first detected in Wuhan, Hebei Province, China [1]. On 11 March 2020, the World Health Organisation (WHO) declared the COVID-19 virus a world pandemic [2]. This pandemic is considered the worst event since World War II and has affected every part of human life as we know it. Not only were healthcare systems put under strain, but every part of human life and the economies of countries were affected by the world-wide lockdown that was implemented [3]. In an attempt to stop the spread of the virus, various countries closed educational institutions and resorted to immediate online teaching. It can be said that universities had to implement emergency remote teaching (ERT), a newly created term to distinguish it from well-designed and long-term online teaching [4].

## 2. Literature Review

The physical closure of higher education institutions (HEIs) affected over 220 million tertiary students [5]. Even though the closure proved to be an effective strategy in breaking the transmission chain in the pandemic [3], it still had various negative effects on students' studies [3]. In an attempt to keep the disruption of studies to a minimum, various HEIs across the world implemented online learning programmes by converting face-to-face programmes [3]. The decision to close tertiary institutions was made in order to ensure that, while students would still be able to continue with their studies, they would also be kept safe from possibly contracting COVID-19 [6]. The quick move to online teaching forced many instructors to change their teaching pedagogies and to continuously adjust and adapt to changes taking place [6]. Universities were also forced to modify their assessment policies to address the new methods of assessment administration and proctoring [7].

Research has shown that students found the transition very challenging [3]. The forced move to online education has exposed various emerging vulnerabilities in education systems worldwide [8].

*Challenges of Online Learning*

Online learning can be defined as learning that uses Internet networks, providing accessibility, connectivity, flexibility, and the ability to produce various types of learning interactions [9]; under normal circumstances, it would take six to nine months to properly plan, prepare, and develop content for online courses [10]. Owing to the quick, almost emergency transition from face-to-face to online teaching, concerns have been raised about the quality of the curriculum, instruction, and assessment practices [11]. The majority of multi-campus universities make use of a learning management system (LMS) to facilitate class attendance, student registration, and content distribution, but not for fully online teaching/classes [1]. Various countries have made significant attempts to ensure the continuance of teaching, whether using radio and television or strengthening their current e-learning platforms [1]. Despite several infrastructure limitations, countries such as Italy, South Africa, and Australia have moved to synchronous and asynchronous learning [1].

The abrupt change from contact classes to online teaching brought many challenges. Some of the challenges experienced are indicated in Table 1 below:

**Table 1.** Challenges experienced in other countries.

| Challenge | Explanation |
| --- | --- |
| Electricity | In India, there are still parts where power has not yet been restored after cyclone Amphan destroyed the electricity infrastructure. This made it difficult for students and lecturers in these areas to charge their devices and work online [12]. |
| Internet Connection | Not all students have access to broadband/Wi-Fi connections and most make use of cellphone data. This has a negative impact on the online teaching experience and also affects the speed with which material can be downloaded and assignments uploaded [12]. Various developing countries still struggle to provide reliable and stable Internet connections, which can lead to students missing submission deadlines [1,13]. The majority of students who studied on campus before COVID now had to study from home, with no or very basic Internet connectivity [6]. Internet connectivity not only affected students, but also lecturers, who had to consider elements such as Internet speed to ensure classes could be held online in an effective manner [11]. |
| Devices | The majority of students do not have access to a full range of devices, such as laptops and tablets/iPads. Students use their smartphones to do their online learning, which makes it very difficult to read the learning content properly [7]. Students in various countries that are economically struggling may find it difficult to afford more than one online device [1,13]. Students' devices might not be upgraded enough to handle the applications required for online courses [11]. |
| Disturbances | Students do not always have their own room/study area where they can sit and attend online classes/pay attention to their studies, due to families living in small houses and siblings that are at home. This provides a big challenge for students to concentrate on their studies. The disturbance of siblings playing in the background/family members having a conversation is not only disturbing to the student, but also to the lecturer trying to conduct a class [12]. Since the daily routine of going to university had been disrupted and the entire family was home, students did not feel the urgency to follow a schedule and complete their studies; they would rather spend time on relaxing activities [13]. |

**Table 1.** *Cont.*

| Challenge | Explanation |
|---|---|
| **Dealing with stress** | The rapid change from face-to-face to online learning increased stress for various stakeholders at universities. IT departments had to integrate existing external applications into their current systems and had to ensure that IT infrastructure could cope with the increased online presence of academic staff members and students [8,13]. Staff members experienced increased stress levels as they were not properly trained in all the applications that they now needed to use for lecturing [10], and they had to transform the content from face-to-face delivery to online delivery in a very short time, which could result in increased financial and time constraints [13]. |
| **Lack of human contact and collaboration** | It is difficult to maintain active participation with students in an online environment where you cannot make eye contact with students. It also makes it very difficult for practical classes to be facilitated. The risk of not having eye contact with students during a lecture is that students can become passive, and this can result in negative outcomes [12]. Online teaching creates a further limitation in that practical work cannot be taught and students reported that it was difficult to pay attention during online lectures [14]. Students, especially those with low self-confidence, can feel excluded from the learning group very quickly [15]. |
| **Quick transitioning to online learning** | Faculties were faced with various challenges regarding the change from face-to-face teaching to online teaching. Instructors had to make use of LMSs in which they were not properly trained to deliver content, and still had to ensure effective learning was taking place [11]. Content used for face-to-face lectures had to be re-worked to be applicable for online teaching [11]. Educators felt intimidated by online learning due to their inadequate technological pedagogical knowledge and limited knowledge regarding online teaching [14]. |

## 3. Research Methodology

Research design is the method used for finding answers to your research questions as accurately, objectively, and economically as possible [16]. Research design provides a method for a researcher to address a research problem in a logical manner [17]. The research design selected for this research was a qualitative design with an interpretive paradigm as the framework. A qualitative research design can be defined as an approach whereby research findings are reached without making use of statistical measures [18]. This is emphasised by Harris, Gleason, Sheean, Boushey, Beto, and Bruemmer [19] when they state that qualitative research can provide findings that are not of a quantitative nature, such as numbers, but rather text, audio, or visual findings. Qualitative research was suitable for this study as, according to Creswell [20], it was used to achieve a perceptive knowledge of a distinctive establishment or occasion and the exploration of the content with participants at their locations. Sequential analysis can be defined as " ... projects conducted one after another to further inquiry, with the first project informing the nature of the second project" [21]. The information obtained from the questionnaire in the qualitative research was used as a base for mathematical calculations to incorporate the quantitative element of the findings and to enhance the findings with a deeper understanding. Quantitative research can be defined as "a set of strategies, techniques and assumptions used to study psychological, social and economic processes through the exploration of numeric patterns" [22]. This method is used to find patterns and averages, make predictions, test causal relationships, and generalise results to wider populations [23]. From the above definitions, it is clear that quantitative research refers to the numerical element that is applied to determine results.

### 3.1. Research Sample

To perform the research, two multi-campus residential universities in South Africa with similar demographics and student populations were selected. These universities were selected as their student bodies are similar in terms of demographic and also in terms of

student circumstances. The majority of students at the selected institutions come from previously disadvantaged backgrounds and are first generation students. A multi-campus university can be defined as an institution where campuses are geographically separated from each other but combined in a single university system [24]. Scott, Grebennekov, and Johnston [25] further define multi-campus universities as institutions with multiple campuses in multiple locations, consisting of three or more campuses, and with more than 60% of the total student body on the main campus. Residential universities can be defined as institutions where contact classes and on-campus living are promoted and the focus is on holistic, contact learning [26]. The universities selected offer both degree and diploma courses and undergraduate and postgraduate courses for accounting students. The participants in this study were the lecturers who lecture on diploma, undergraduate and post graduate level for accounting degrees. For this study, the researchers applied the purposive sampling technique to identify the respective universities. Purposive sampling means that the researchers select participants for a study because they can enlighten an understanding of the research predicament "purposefully" and to identify significant phenomena in the study [20]. In other words, the researchers decide what needs to be known and set out to find people who are willing to provide the information by virtue of knowledge or experience [27]. Purposive sampling typically involves identifying and selecting individuals who are especially knowledgeable about or have experience of a phenomenon of interest [28]. The purposive sampling method was an advantage to this study, as it was an accurate representation of the sample universities.

A self-developed questionnaire was used as a research tool to obtain the relevant information for the study. Confidentiality was ensured by not disclosing the participants' details and the relevant ethical clearance was obtained. The questionnaire was sent out to all lecturers in the accounting departments of the selected two universities, and 70 lecturers responded.

The research was based on the stakeholder theory, as the actions of a university can affect various stakeholders, such as students, alumni, staff, industries and professional bodies associated with the university, and the government [29]. The stakeholder theory is different from the shareholder theory in the sense that it focuses on all stakeholders of an organisation [30]. For this reason, the stakeholder theory was the basis for the study.

The stakeholder theory also fits in with the idea of doing business ethically. Business ethics can be defined as "the values and standards that determine the interaction between business and its stakeholders" [31]. This definition further ties in with the definition of ethics, which can be defined as good, self and other—something is ethical when it is good for oneself and others [31].

### 3.2. Limitations

This study had some limitations since it focused only on multi-campus universities, and it only focused on two universities that had similar student bodies, where the majority of the students are first generation students and secondly previously disadvantaged students. The type of qualifications compared are similar in the two universities.

## 4. Results and Discussion

A questionnaire divided into two parts was used. The objective of the first part of the questionnaire was to obtain general information about the level at which lecturers were lecturing, whether it was a degree or a diploma, and the subject lectured. The second part focused more on how assessments were written and the challenges that were faced, whilst ensuring the integrity of the written assessments.

### 4.1. General Information

i.    Findings

The majority of the respondents (72%) were lecturers for an accounting degree. Of the 72%, 33% lectured at a third-year level, which is an exit level at both institutions and

requires that students would be proficient to a certain standard to obtain their degrees. Of the 33% lecturing at a third-year level, 57% lectured in auditing.

*4.2. Conversion from Traditional Methods to Online*

i.      Objective

The objective of this question was to determine whether specific year groups or subject groups struggled more with conversion from a face-to-face method to online than others.

ii.      Findings (see Figure 1)

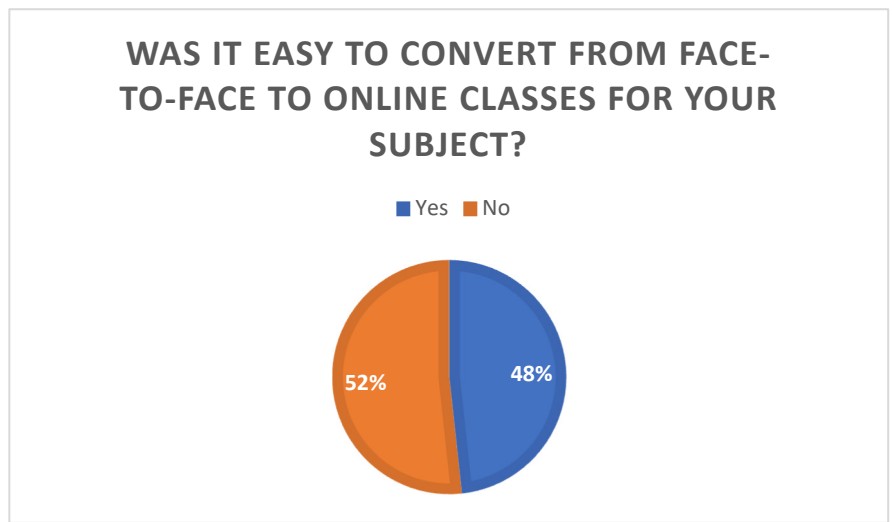

**Figure 1.** Was it easy to convert from face-to-face to online classes. (Source: own analysis).

From the results above, it is evident that the answers were almost tied in terms of the ease to convert. Only 3% more respondents indicated that it was difficult to convert. From the respondents that indicated it was difficult to convert, 53% lectured at third-year level and 37.5% lectured in financial management. Furthermore, the results indicate that lecturers for the accounting degree had struggled more to convert to online teaching than lecturers at the diploma level. It can be concluded that subjects that are more complex and that require a higher level of understanding and explanation regarding difficult topics had been more difficult to convert than those where standards and legislation were available to assist in explaining difficult topics. Some of the reasons given for struggling to convert included, but were not limited to:

- Lecturers found themselves in unfamiliar territory, which created anxiety;
- There was not enough time for proper preparation and conversion;
- Lecturing styles had to be reinvented in a very short time to ensure students remained interested during lectures;
- Lecturers were unsure how to incorporate technology as a lecturing method and how to record videos.

Some of the comments received from the participants included:

"A number of challenges existed that needed to be overcome mainly regarding upskilling oneself with the technologies that support an online platform of tuition. Blackboard had to be mastered as the engine to both provide as well as assess the accounting module my colleague and I offered to 1st year students. Students had a host of challenges relating in the main to the unfamiliarity of the method of instruction done electronically by way of Blackboard scheduled classes. A great deal of time was also expended seeking out relevant supporting material that my colleague and I uploaded onto Blackboard. Added to this was the problem that not all students had laptops either and data was to some extent limited

too. Students also needed to deal with power outages and network connectivity issues too."

"I was concerned about student engagement and interaction. I do not know if students understand a concept or not because they hardly ask questions, but you can easily see on their faces when they don't understand."

Of the 48% of respondents that felt the transition went smoothly, 33% lectured at a second year level (80% on the accounting degree) and 33% at the honours level. There was no specific subject that stood out from the respondents that felt the transition was easy; all subjects were represented to some extent. Some of the reasons why the transition was perceived as easy included, but were not limited to:

- Some lecturers were already involved in developing online modules prior to COVID-19 so they knew what was expected and how to get it done.
- Some lecturers had been offering online classes before COVID-19 and were thus comfortable with the use of technology.
- Lecturers embraced the challenge to rethink the way in which they were teaching.
- Being able to use Blackboard (Bb) as a method to do online teaching assisted with the transition, as lecturers were comfortable with it, although they did not know of all the available opportunities that it could offer.

"Our Department leadership was decisive and communicated transparently. Support was offered. The Department had also built a recording studio in the year before COVID which had been utilised to make video's as supportive material before COVID, so the transition was easier."

"The university had access to the necessary technological infrastructure to allow a fairly easy transition to online learning. The University Learner Management System assisted us in making all the content available in a systematic manner. We could also use the LMS to communicate to students. We furthermore acquired webinar software licenses that assisted us in facilitating larger interactive class sessions and discussions."

"I enjoyed the interaction with the students. The students felt more at ease to ask questions anonymously. I was able to run polls to gauge the understanding of the class rather than a general groan/yes in person. Less distractions as students were not making a noise (muted). Able to offer more value through pre-recorded lectures and additional workshops that would not be offered in a face-to-face environment due to time and venue constraints."

Looking at the above results it seems that the level at which the lecturers taught, as well as the content of the subjects, had a role to play in whether the transition was easy or not. However, it is also clear that the exposure lecturers had to technology, the online environment, and the development of online courses prior to COVID-19 had a significant impact on their ability to convert from traditional methods of teaching to online teaching. From the above, it can be concluded that the lecturers who lectured on more complicated subjects, and who did not have prior exposure to online teaching, experienced the transition as being more challenging, which is in line with the literature that states educators felt intimidated by online learning due to their inadequate technological pedagogical knowledge and limited knowledge regarding online teaching [32].

### 4.3. Challenges Experienced

i.    Objective

The objective of this question was to determine what challenges students faced during this time and how they were overcome.

ii.    Findings (see Figure 2)

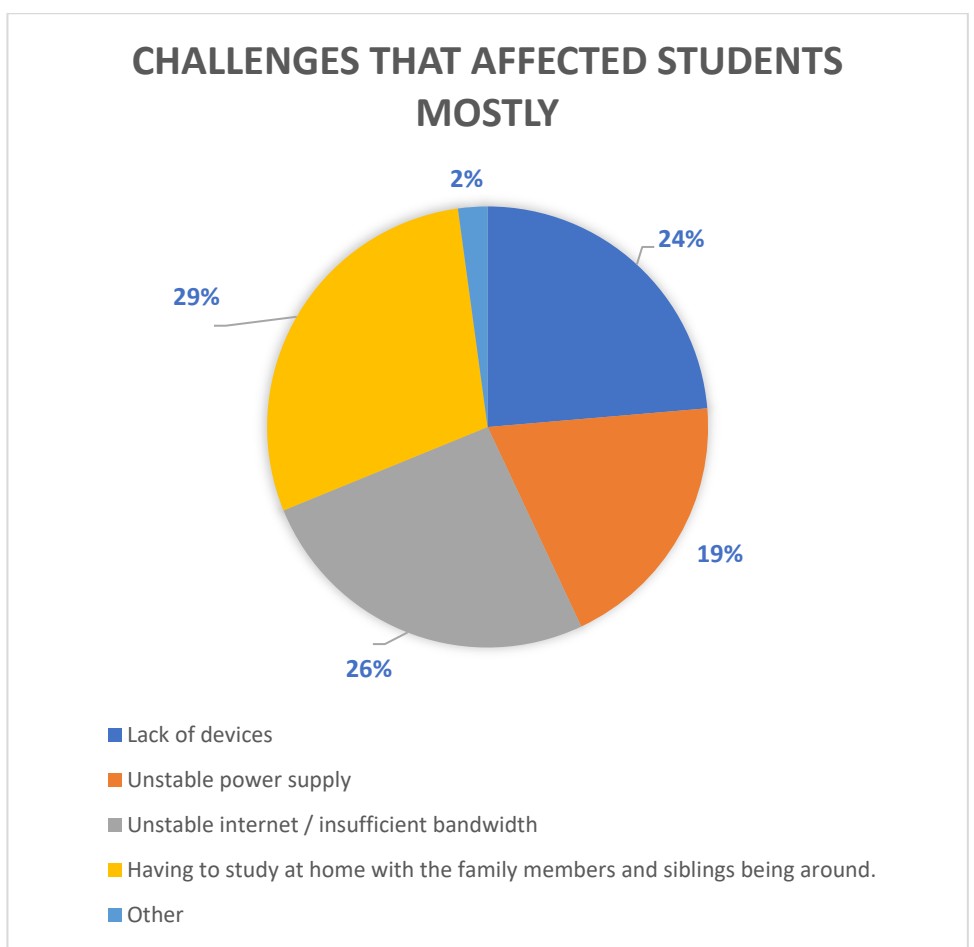

**Figure 2.** Challenges that affected students mostly (Source: own analysis).

The results indicate that the biggest challenge students faced was having to continue with their studies, attending online classes and preparing and writing assessments, while the entire family was at home. Not all government schools were in a position to ensure children could work while not at school, which increased the challenge that university students faced during the COVID period—younger siblings might be playing and making a noise, while students had to focus on their studies. Parents were also home, especially during the hard lockdown at the beginning of 2020, and they were busy with their own things around the house, which could disturb the quiet environment available on campus that students had become used to when they started their studies.

The second biggest challenge faced was stability of the Internet. South Africa is not known for Internet stability. Even when fibre is installed, there is no guarantee that students will have enough bandwidth or an adequately stable Internet connection to attend an entire lecture online. This had been a challenge for students, especially in the rural areas, where they are dependent on data on their mobile devices or the use of dongles. Some students had specific data bundles, which included specified night owl data, which limited students to specific times to download documents and lecture recordings, meaning they were not able to join in live sessions and to ask questions.

It was expected that most students would struggle with not having devices, but the research indicates that only 24% regarded the lack of devices to be a challenge. While investigating this further, it was found that those students who did not have a laptop, PC, or tablet had obtained a smart phone (not always the latest model) and had used the phone to download documents, read slides, and to study with. They still had a challenge

in uploading assignments, which meant most students would write out the assignment instead of typing it, take photos of their attempts, and email it to the lecturer or upload the photos via their phones on Blackboard.

Lack of a consistent power supply is something South Africans have learned to live with. Only 19% of the respondents indicated that this had been a challenge. Most of the time, individuals had learned to manage their activities and rearrange their plans in accordance with load shedding schedules. Sometimes it is not possible, but from the above it seems as though this was the best-managed challenge faced by respondents.

Some of the comments received included:

- Students' inability to motivate themselves to study due to the presence of their families;
- Students tended to experience depression due to a lack of social interaction with their peers;
- Inability to plan. Students were used to continued reminders like deadlines and submission dates, which was not the case during the online learning;
- Although students made use of various devices, they were not always sufficiently trained on how to use the devices for academic purposes;
- Managing stress levels. Students were stressed about finances and their ability to continue to study, as many parents lost their jobs, and this impacted on students' ability to focus on their studies.

Some of the feedback provided below was obtained through open-ended questions included in the questionnaire:

"If a student was not naturally VERY self-motivated and self-disciplined, the impact of this was exponential during COVID. Self-driven students succeed anyway. Students who did not know how to manage everything and adapt, did not."

"Students are very unfamiliar with research, and therefore, research orientated modules are often a daunting and lonely process for students. In this light, face-to-face classes provide students with a sense of community and make the research process feel less isolated and lonely. So, I think the move to online classes exacerbated the feeling of research being a lonely process."

"Socio-economic problems at home were compounded during COVID. So, if a student already came from a poor family, the COVID restrictions made them even poorer. Also, some students and their families became very sick during COVID, and some students lost friends and family, which added to their stress and worries about the future."

In order to help students to cope with the online learning environment, lecturers and the university have had to attempt to address as many of these challenges as possible. The following methods were implemented by the universities:

- Students were provided with data bundles to assist them in having data during the daytime to participate in online lecturing activities.
- The number of attempts to submit assignments were increased and the submission times were extended to accommodate students struggling with connectivity issues or being affected by load shedding.
- Students on NSFAS bursaries were provided with devices so that they could continue with their studies.
- The Centre for Academic Technologies (CAT) provided training to all students on how to approach the online learning environment.

Some solutions implemented by the lecturers:

- Provided bigger picture thinking and planning to the students to help them get through the work.

- Students at risk of not completing their modules, due to various reasons, were continually identified and regular meetings were set up by their tutors to try and assist as far as possible.
- Various chat groups were started on platforms such as WhatsApp and Nova, where students could raise concerns and connect with each other.
- Weekly guidance documents were drawn up to ensure students knew exactly what they had to do on a weekly basis.
- Videos were recorded of all the lectures and uploaded for students to download and watch at a convenient time for them.
- The lecture times were shortened, and videos were limited in size to allow for ease of download.

From the above, it is evident that the universities provided many students with devices and data to assist with the challenges of data and stable Internet connections. Lecturers also assisted students by shortening lecture times and limiting video lengths and sizes, as well as recording lectures, so students could access them at convenient times. Lecturers also accommodated students with difficult circumstances at home by allowing multiple submission attempts. The empirical results confirmed that some of the challenges experienced by South African multi-campus university students were similar to what the rest of the world had experienced, such as inconsistent and slow Internet access [1,6,13], and a lack of a consistent power supply, which was shared with India [12], although the causes were different. Another challenge shared with international countries was devices. Devices were either too old to handle the online content, or students did not have devices like iPads/tablets/computers, etc., and had to do everything on their smartphones, which hindered the learning experience [11,12]. The last challenge that correlates to challenges experienced in other countries is the fact that students had to study from home, where siblings and parents were present, when they were used to a campus environment conducive to studying [3,12].

### 4.4. Challenges Faced by Lecturers

i.    Objective

The objective of this question was to determine lecturers' biggest challenge during the lockdown period, as well as whether there are similar challenges experienced by the lecturers as they observed from the students.

ii.    Findings (see Figure 3)

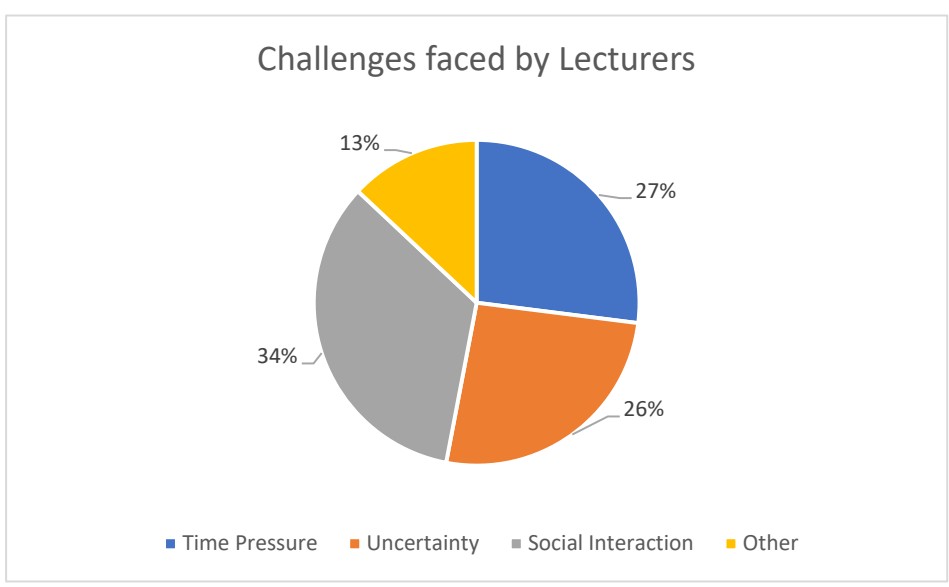

**Figure 3.** Challenges faced by lecturers. (Source: own analysis).

The findings indicate that three significant challenges stood out, which will be discussed below. However, none of the challenges experienced by the lecturers corresponds with the challenges the students experienced.

*Social Interaction*

The biggest challenge was a lack of interaction with the students (34%). Lecturers were not able to form a relationship with their students, like they had prior to COVID-19. Owing to the online delivery of content, classes were felt to be very mechanical, and the emotional element associated with lecturing was absent. There was no immediate contact with students to assist with small issues regarding academic content. This correlates with the literature review, where it was noted that the lack of engagement with students in class complicated lecturers' ability to gauge students' level of understanding and made it difficult to ensure participation took place during lectures [12].

*Time Pressure*

The second challenge was time pressure (27%). Lecturers struggled to manage their time in terms of learning the new technology and applications that had to be used in the limited amount of time, while still having to convert content and keep up to date with the normal workload. Students obtained some lecturers' cellphone numbers, and this led to lecturers feeling they were constantly working due to students messaging them at all hours of the day.

*Uncertainty*

The third biggest challenge was uncertainty (26%). Lecturers were uncertain if the students followed the guidance documents given to them and if they really stayed on track with the learning programme. They were also uncertain how much the students took in during lectures as they struggled to keep them engaged. It was difficult to gauge their level of understanding during an online lecture, when students' cameras were turned off to save bandwidth.

*Other*

Other challenges include difficulty to manage the new expectation levels, difficulty managing discipline in the 'classroom' and getting use to the 'mechanical feel' of the online classroom. Lastly, some lecturers indicated that the online teaching environment felt very lonely and that they struggled to adjust to it.

*4.5. Assistance from Employer to Convert to Online Learning*

i.    Objective

The objective of this question was to determine whether sufficient support was given by the various employers to assist staff members to convert from traditional learning methods to online teaching.

ii.    Findings (see Figure 4)

The majority of respondents (66%) indicated that sufficient support had been provided by their institutions to assist with the transition to online learning. Some of the methods that were used to assist lecturers included but were not limited to:

- Teaching and learning workshops were run to assist in conducting online learning, including the different uses of Blackboard in these circumstances.
- Having support from members of the department at various levels to assist with any issues.
- During the transition, management became aware of the various needs of students and lecturers and their support increased.
- Different year groups came together to determine how the online environment could be managed for their group and degree; whether by using roadmaps or employing small milestone submissions.

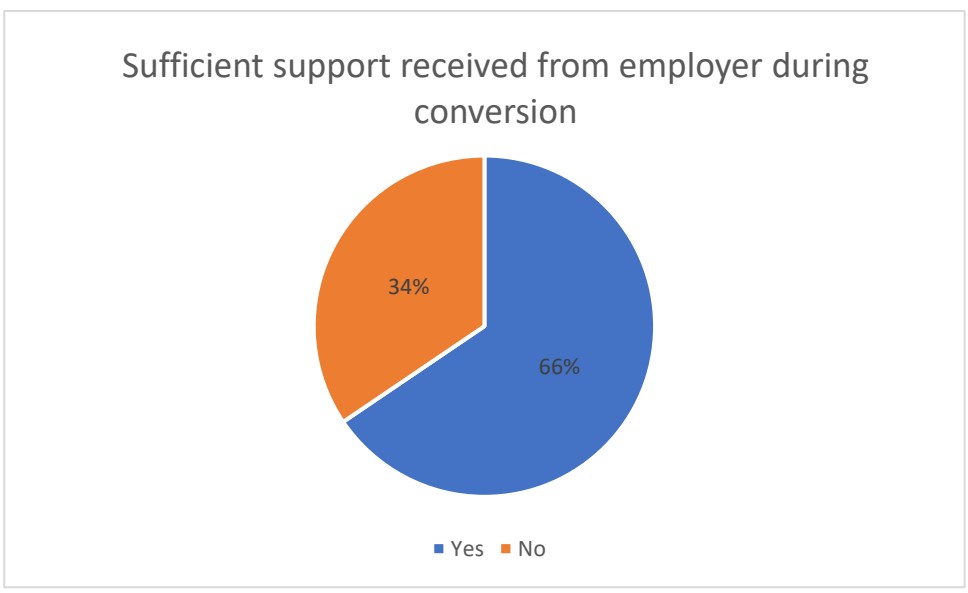

**Figure 4.** Sufficient support received. (Source: own analysis).

Some of the comments included:

"My answer above is YES and NO. Training was provided, and recommendation was provided but, in the end, it came down to the lecturer to come up with good ideas to make this work in a very short amount of time."

"Employer provided assistance through university platforms and Departmentally. In addition, departmentally we had forums to share our experiences, so that we could create best practices for online teaching and learning."

The respondents who indicated that no training had been provided felt that time was a big problem, since the transition was taking place very quickly with insufficient time to obtain proper training. They had to train themselves, as the provided training was very generic and not addressed to their needs. Lecturers also felt that most things were done reactively rather than proactively. One of the comments encapsulates the experience:

"Useless and irrelevant training. No best practice shared amongst the accounting years with every year doing different things in terms of assessments. No support from management."

## 5. Conclusions

The purpose of this paper was to determine the challenges students and lecturers faced at multi-campus universities during the move to online learning. Lecturers from two multi-campus universities were asked to provide their experiences of the challenges they faced, as well as their awareness of the challenges faced by students. Historically, these institutions offered qualifications such as diplomas and accounting degrees, but primarily according to the traditional contact teaching method.

The literature indicates that the pandemic forced universities to move from traditional face-to-face lectures to full online lecturing in a very short time, with most universities not having a sufficient infrastructure and students not being properly prepared for the change. Even though most of the learning management systems used by universities were able to deal with assignment submissions and class attendance registers, most of these systems were not built for full online teaching and learning.

The literature further indicates that the biggest challenges that universities faced with the integrity of assessments were lack of electricity, lack of data, increased stress levels, and changes in home circumstances.

The empirical findings indicated that similar challenges had been identified in South Africa. The biggest challenge faced by South African students, from the viewpoint of lecturers, was the disruption of their day-to-day lives, especially for students who were not able to motivate themselves to continue with their studies. Students also struggled with proper Internet connectivity to ensure they stayed on course with their studies and attended classes. The lack of guidance and lack of social interaction with their peers increased their stress levels, and some students also showed the beginning stages of depression. The issue of students not having sufficient devices to properly engage with their studies was dealt with by each country individually. In South Africa, this was resolved by providing devices to students who could not afford their own, and universities also provided all students with a significant amount of data to assist in attending lectures and the downloading of content. Although various attempts were made to assist students in overcoming these challenges, some challenges, such as circumstances at home and loadshedding, were out of the hands of the institutions and lecturers.

Lecturers were also faced with challenges in the move to online learning, since sufficient and appropriate training was not always available, and lecturers had to devise creative ways to deliver online content on their own. They also had to ensure that they managed their time properly, as very little time was available for the transition. It would seem that lecturers who were involved in the delivering of online modules prior to the pandemic had a more seamless experience of the transition than those who had not.

Areas for future research include investigating the impact blended learning has had on students' ability to cope with change and investigating the best methods to assist students in bridging the gap between continued assistance and thinking on their own.

The study is of value as it provides valuable insight about challenges faced by multi-campus universities in South Africa, in comparison to other countries, when traditional classroom teaching was disrupted during COVID-19.

**Author Contributions:** Conceptualization, C.H.-d.P. and B.M.; methodology, C.H.-d.P.; formal analysis C.H.-d.P.; investigation, C.H.-d.P.; resources, C.H.-d.P.; writing—original draft preparation, C.H.-d.P.; writing—review and editing, B.M.; project administration, C.H.-d.P. All authors have read and agreed to the published version of the manuscript.

**Funding:** This research received no external funding.

**Institutional Review Board Statement:** The study was conducted in accordance with the Declaration of Helsinki, and approved by The School of Accounting Research Ethics Committee (SAREC) at the University of Johannesburg. SAREC20220623/06.

**Informed Consent Statement:** Informed consent was obtained from all subjects involved in the study.

**Data Availability Statement:** Due to ethical reasons and conditions under which ethical clearance was granted, the data used for this study cannot be made available.

**Conflicts of Interest:** The researchers are employed at one of the institutions that took part in the study, however the questionnaire was distributed by the relevant HoD's of the departments and the responses were all anonymous to ensure the integrity of the answers.

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
