# Peer review of "Challenges Faced by Multi-Campus Institutions with Online Teaching during the COVID-19 Lockdown"

_education, doi:10.3390/educsci13040419_

Round 1

Reviewer 1 Report

The paper is devoted to the current topic connected with educational process during the Covid-19 pandemic situation. The change from face-to-face to online learning is discussed from the consequences for students and lecturers on various levels. The determination how challenges faced by multi-campus universities in South Africa during this pandemic time is discussed and compare also with other countries. Questionnaire research method was used. Interview with participants brings interesting description how the Covid-19 situation changed their study.

I have some recommendations:

1-      There is not needed to write in the abstract so long about general situation during the pandemic time. This part can be shorter.  I recommend putting some information about research sample of questionnaire. Which participant were asked? Can you characterize the study programme and put the name of the university?

2-    On the page 2 after the sentence “Some of the challenges experienced include: “ is one part with another typesetting. Is it one table? I think it is possible to continue with normal text and to use bullets.

3-      On the page 3 is the part, which start with the sentence „Research design is the method used... In the case, that the authors wish to present their research methodology, I recommend adding her the name of the paragraph with title “Research methodology”. It is the possibility here to describe questions in the questionnaire. It is written that the authors used mixed method design, but we think, that it was used more qualitive pedagogical research, that presented some results in the form of descriptive statistics with percentage. It is enough to describe only methods used in the research, another not used methods disturb the understanding of the text.  

4-      On the page 4 is the part, which start with the sentence „To perform the research, two multi-campus residential universities in South Africa, with similar demographics and student population, were selected. In the case, that the authors wish to present their research sample, I recommend adding her the name of the paragraph with title “Research sample”.

- It will be better for understanding for the reader, if authors in the parts 3.1, 3.2 and 3.3 Findings add more own statements to the presented results and compare them with scientific literature or research studies.

- Some information about research sample is missing. How many students were asked in the questionnaire research? Which kind of study they attend (bachelor, master, teacher training, electrical engineering, …)?

5-     On the page 9 is written “specifiaclly shared with India  “. It should be written “specifically” . Why is comparing with India in this place?

6-     They are on the page 5, 7, 9 and 10 colour figures. It is possible to add description to them – Figure 1, Figure 2,…, Figure 4. It is expected in this case to cite figures in the text of the paper.

 -   In the Conclusion are described the meanings of lectures and students about their experiences with teaching process during pandemic situation. It is not clear in the text of the paper, which was the difference between research with students and research with lectures. They both obtain the questionnaire? They obtain questionnaire with the same questions? In the conclusions is also written “The lack of devices was solved in various ways by various countries.“ This sentence is too broad, it is needed the specification here.

8-    The research has some limitations. These limitations of the realised research is missing in the paper. 

Reviewer 2 Report

Please see all the specific and detailed comments in the attached file.

Several general suggestions:

·       Restructure/reorganize the second section – blue inked text (what the authors did) is very “diluted” in the middle of so specific citations and hard to follow.

·       Insert the number of participants in your research - this is never mentioned.

·       Avoid the indirect citations – too many (the oldest one of a 2002 document) and three of these references were only used once for the corresponding indirect citation.

·       Pictures should be changed, removing the questions (at least as big as these are) and legends should be inserted.

·       Please clearly explain how the lectures comments that are cited in the text were obtained (by the open-ended questions? Interviews?...).

·       Some tables with crossed values would be nice to see – these could be appealing for the reader.

Since there is no mention to any questionnaire for students, the corresponding results presented must be clearly mentioned as “from the lecturers’ perceptions”, in other words, only one questionnaire is mentioned and it regards only lecturers (last paragraph of blue inked text - Article commented PDF file) – several results are pointed regarding students but are not mentioned “as lecturers’ perceptions of…”.

Rethink the conclusion section, at least in what students is concerned.

See also:

·       Several missing in text citations in the reference list (red inked in the text).

·       References in reference list not mentioned in the text;

·       Improve the references (all commented).

Round 2

Reviewer 2 Report

Several points were clarified and corrected, which is positive!

However:

- Once again, an image changing suggestion: remove the "questions" from the image space and place it in the corresponding legend - this can clarify the "image sense" and impact.

- The "objective/information" regarding Fig. 3 is still quite "empty" since it transmits a sequence like "1 leads to 2"; "2 leads to 3"... however this, as I see it, should transmit to the reader the most impactive challenges - not a sequence...

Some references still need to some corrections, for instance:

[2] "  Babbar, M. and Gupta, T. Response of educational institutions to COVID-19 pandemic: An inter-country comparison. Policy 449 Futures in Education 2021, p.14782103211021937. "

(?)... what is this number?... Please insert, after the year 2021 (despite the 1st publication - recommended reference being from 2022). 20(4), 469–491. doi: 10.1177/1478210321102193

[6] and [7] - I really think authors are mentioning Creswell, not Cresswell - please correct

[25] Sando ... this does not seem correct - please (once again) see if this should be changed to: Fiano, K.S, Medina, M.S. & Whalen, K. The Need for New Guidelines and Training for Remote/Online Testing and Proctoring. Am J Pharm Educ. 2021 Sep;85(8):8545. doi: 10.5688/ajpe8545.

[31] ? is this reference needed?... it seems out the context of this paper and it is used just for a presented "definition" of Residential Universities (unique online citation)...

Author Response

Reviewers comments

Response

- Once again, an image changing suggestion: remove the "questions" from the image space and place it in the corresponding legend - this can clarify the "image sense" and impact.

Legends have been inserted in all the graphs provided.  The first graphs cannot have a different title as then the yes and no in the legend will not make sense.

- The "objective/information" regarding Fig. 3 is still quite "empty" since it transmits a sequence like "1 leads to 2"; "2 leads to 3"... however this, as I see it, should transmit to the reader the most impactive challenges - not a sequence..

This has been changed to include a graph like the rest and the order in which the challenges are addressed have been changed as well.

Some references still need to some corrections, for instance:

[2] "  Babbar, M. and Gupta, T. Response of educational institutions to COVID-19 pandemic: An inter-country comparison. Policy 449 Futures in Education 2021, p.14782103211021937. "

(?)... what is this number?... Please insert, after the year 2021 (despite the 1st publication - recommended reference being from 2022). 20(4), 469–491. doi: 10.1177/1478210321102193

Reference has been updated with the correct information

[6] and [7] - I really think authors are mentioning Creswell, not Cresswell - please correct

This has been updated

[25] Sando ... this does not seem correct - please (once again) see if this should be changed to: Fiano, K.S, Medina, M.S. & Whalen, K. The Need for New Guidelines and Training for Remote/Online Testing and Proctoring. Am J Pharm Educ. 2021 Sep;85(8):8545. doi: 10.5688/ajpe8545.

This is the exact same article with the same people, however I’ve inserted the online journal information.  The reference originally included was the PDF document downloaded and there the name appears as Karen Sando.

[31] ? is this reference needed?... it seems out the context of this paper and it is used just for a presented "definition" of Residential Universities (unique online citation)...

Yes, as it is a definition used that is not mine.  Therefore required in the reference list.
